# High-affinity interactions and signal transduction between Aβ oligomers and TREM2

Christian B Lessard[1], Samuel L Malnik[1], Yingyue Zhou[2], Thomas B Ladd[1], Pedro E Cruz[1], Yong Ran[1] (iD),
Thomas E Mahan[3], Paramita Chakrabaty[1,4], David M Holtzman[3] (iD), Jason D Ulrich[3], Marco Colonna[2] &
Todd E Golde[1,4,*] (iD)

## Abstract

Rare coding variants in the triggering receptor expressed on myeloid cells 2 (TREM2) are associated with increased risk for Alzheimer's disease (AD), but how they confer this risk remains uncertain. We assessed binding of TREM2, AD-associated TREM2 variants to various forms of Aβ and APOE in multiple assays. TREM2 interacts directly with various forms of Aβ, with highest affinity interactions observed between TREM2 and soluble Aβ42 oligomers. High-affinity binding of TREM2 to Aβ oligomers is characterized by very slow dissociation. Pre-incubation with Aβ is shown to block the interaction of APOE. In cellular assays, AD-associated variants of TREM2 reduced the amount of Aβ42 internalized, and in NFAT assay, the R47H and R62H variants decreased NFAT signaling activity in response to Aβ42. These studies demonstrate i) a high-affinity interaction between TREM2 and Aβ oligomers that can block interaction with another TREM2 ligand and ii) that AD-associated TREM2 variants bind Aβ with equivalent affinity but show loss of function in terms of signaling and Aβ internalization.

**Keywords** Alzheimer's disease; amyloid; APOE; innate immune response; TREM2

**Subject Categories** Genetics, Gene Therapy & Genetic Disease; Neuroscience

## Introduction

The most prevalent form of dementia, Alzheimer's disease (AD), is hypothesized to be triggered by accumulation of aggregated amyloid-β (Aβ) followed by a "cascade-like" chain of events that includes induction of tauopathy, neurodegeneration, and alterations in innate immune signaling (Hardy & Selkoe, 2002; Musiek & Holtzman, 2015). The later feature is reflected by the presence of a reactive astrocytosis, microgliosis, and increased level and accumulation of a variety of immune molecules. A more central role of the innate immune system and microglial cells in particular has emerged from genetic studies. These studies show that numerous loci encoding immune genes are associated with altered risk for AD. Even more compelling are data showing that coding variants in three different genes (*TREM2, PLCG2,* and *ABI3*), whose transcripts are expressed primarily in microglial cells in the brain, alter risk for AD (Golde *et al*, 2013; Jin *et al*, 2014; Sims *et al*, 2017). Despite these associations between the immune system and AD, there is little consensus as to how alterations in the immune system mechanistically alter AD risk. Modeling studies reveal a complex relationship between immune activation states, the proteinopathies found in AD, and neurodegeneration—a phenomenon we refer to as immunoproteostasis (Chakrabarty *et al*, 2015). Further, definitive insight into how genetic variations within immune loci that alter AD risk alter immune function has remained enigmatic. Further understanding of how genetic risk factors impact the immune function will be important to guide therapeutic development, that to date has largely focused on anti-inflammatory strategies (Chakrabarty *et al*, 2015).

The rare protein coding variants in TREM2, R47H, and R62H, which are reproducibly associated with increased risk for developing AD, are being intensively studied as these variants were the first coding variants in an innate immune gene that have been reproducibly shown to alter risk for AD (Cruchaga *et al*, 2013; Jin *et al*, 2015; Lill *et al*, 2015). Prior to association with AD, homozygous or compound heterozygous mutations in TREM2 had been identified in a disease called polycystic lipomembranous osteodysplasia with sclerosing leukoencephalopathy (PLOSL) or Nasu-Hakola disease (Bianchin *et al*, 2006). PLOSL is characterized by fractures, frontal lobe syndrome, and progressive presenile dementia beginning in the fourth decade. Numerous studies of the PLOSL-associated TREM2 variants suggest that these variants are loss of function altering maturation and cell-surface expression of TREM2 (Paloneva *et al*, 2001, 2002). Both structural and cell biology studies of the AD TREM2 variants show they lie on the surface of the TREM2 ectodomain, possibly framing a binding pocket, whereas the PLOSL

1  Center for Translational Research in Neurodegenerative Disease, Department of Neuroscience, University of Florida, Gainesville, FL, USA
2  Department of Pathology and Immunology, Washington University School of Medicine, St. Louis, MO, USA
3  Department of Neurology, Hope Center for Neurological Disorders, Knight ADRC, Washington University School of Medicine, St. Louis, MO, USA
4  McKnight Brain Institute, University of Florida, Gainesville, FL, USA
   *Corresponding author. Tel: +1 352 273 9458; Fax +1 352 294 5060; E-mail: tgolde@ufl.edu

variants are within the core structure (Kleinberger et al, 2014; Kober et al, 2016). Maturation studies demonstrate folding and maturation deficits induced by PLOSL-associated TREM2 variants, whereas the AD variants do not consistently show these same functional impairments. Notably, a DNA polymorphism within a DNase hypersensitive site 5′ of TREM2, rs9357347-C, also associates with reduced AD risk and increased TREML1 and TREM2 levels (Carrasquillo et al, 2017). This variant is associated with decreased risk for AD, which would be consistent with the hypothesis that the coding variants in TREM2 associated with increased AD risk are partial loss of function.

TREM2 is a type I transmembrane glycoprotein that interacts with the transmembrane region of DAP12 (TYROBP) to mediate signaling events through DAP12's immunoreceptor tyrosine-based activation motif (ITAM) domain (Peng et al, 2010). The extracellular, immunoglobulin-like domain V type, domain of TREM2 is glycosylated and shows cell-surface localization. TREM2 is highly expressed in microglia and peripheral macrophages (Colonna & Wang, 2016). TREM2 is one member of a family of receptors that have diverged between mouse and humans. In both mouse and human, the family members are clustered within a single chromosomal region on chromosome 6. In the context of neurodegenerative disease, the other family members have not been intensively studied as they are typically expressed at much lower levels in the brain. Like many type I membrane receptors, TREM2 can undergo ectodomain shedding and subsequent intramembrane cleavage by γ-secretase (Wunderlich et al, 2013). The physiologic implications of these proteolytic cleavages remain unclear, though increased soluble TREM2 has been reported in the CSF of early-stage Alzheimer's disease (Suarez-Calvet et al, 2016), and study of a variant p.H157Y reported to be associated with AD in the Han Chinese population has been shown to increase ectodomain shedding (Jiang et al, 2016a; Schlepckow et al, 2017; Thornton et al, 2017). Enhanced shedding activity from the mutation p.H157Y located in the ectodomain cleavage site leads to a reduced cell-surface TREM2 and phagocytic activity (Schlepckow et al, 2017; Thornton et al, 2017).

TREM2 has been implicated in multiple functions including migration, survival and proliferation, cytokine release, and phagocytosis (Kleinberger et al, 2014; Ulrich & Holtzman, 2016; Yuan et al, 2016). TREM2 appears to be a promiscuous receptor and has been shown to interact with cellular debris, bacteria, anionic phospholipids, nucleic acids, and apolipoprotein E (APOE; Atagi et al, 2015; Bailey et al, 2015; Yeh et al, 2016; Kober & Brett, 2017; Song et al, 2017). Alterations in signaling and binding of TREM2 AD variants compared to wild-type (WT) TREM2 have been reported, though the pathogenic relevance of these alterations is not yet completely clear (Kleinberger et al, 2014; Kober et al, 2016). Mouse modeling studies show increased TREM2 expression in multiple models of neurodegenerative diseases, and TREM2-positive microglial cells appear to surround plaques in APP models (Jiang et al, 2014; Ulrich et al, 2014; Yuan et al, 2016; Jay et al, 2017). TREM2 also shows an altered distribution in microglia surrounding the amyloid plaque, with much of the TREM2 immunoreactivity localized within the plasma membrane of the microglial cells adjacent to the plaque. In TREM2 knockout mice crossed with APP, there are fewer microglial cells surrounding plaques and an increase in plaques with "specular" amyloid morphology as well as more neuritic dystrophy (Wang

et al, 2015, 2016). Such data have led to the hypothesis that TREM2 may regulate the interaction of microglia with plaques and protect other brain cells from toxicity.

In this study, we investigated whether TREM2 and its AD variants interact directly with Aβ. We found no significant changes in the binding affinity between Aβ42 with TREM2 or the AD variants or with their mouse counterpart Trem2, Trem1, and Treml1. Despite this lack of difference in the binding affinity, we observed an attenuated activation of NFAT signaling and a reduced amount of Aβ42 internalized by cells expressing the TREM2 AD variants. We show that TREM2, TREM1 and TREML1 bind APOE and Aβ variably. Our study reveals that the TREM2 and other family members can bind various forms of Aβ and that Aβ oligomers can induce singling events. Further, our data are consistent with the hypothesis that AD risk variants in TREM2 induce a partial loss of function (Song et al, 2018), but do not directly alter binding affinity for either Aβ or APOE.

## Results

### Soluble TREM2-Fc and soluble TREML1-Fc interact with Aβ42 fibrils

The ectodomain of TREM2 interacts with various ligands including bacterial liposaccharides and phospholipids. We explored whether TREM2 might interact with Aβ and whether this interaction might be affected by the AD variants. We also analyzed mouse Trem2 in order to compare its relative binding ability to human TREM2. Initial studies focused on whether TREM2 could interact with fibrillar Aβ42 (fAβ), using an Aβ42 pull-down assay we have previously validated (Chakrabarty et al, 2015). Conditioned media from transiently transfected HEK 293 cells expressing the soluble ectodomains of (i) human TREM2 fused to the Fc domain of human IgG4 (sTREM2-Fc), (ii) AD-associated TREM2 variants (sTREM2-R47HFc, sTREM2-R62H), (iii) mouse TREM2 (sTrem2-FC) or (iv) Fc control was incubated with fAβ. These were spun at 18,000 g to pellet fAβ associated proteins. sTREM2, sTREM2-R47H and sTREM2-R62H were enriched in the fAβ pellet (Figs 1A and EV1A), whereas the Fc control was not detected in the pellet. Mouse sTrem2 also showed binding to Aβ fibrils. Similar pull-down assay between fAβ and full-length TREM2 was conducted using the cleared RIPA lysate from TREM2, TREM2-R47H, and TREM2-R62H transiently transfected into HEK293T cells. Although less efficient, presumably because of detergent present in the lysate, full-length TREM2 and the AD variants were pulled down by fAβ (Figs 1B and EV1B).

Multiple studies show TREM2 ectodomain interacts with APOE (Atagi et al, 2015; Bailey et al, 2015; Yeh et al, 2016). To confirm that the sTREM2-Fc proteins are functional and bind APOE3, we co-transfected TREM2-Fc constructs with APOE3 and pulled down proteins that bound sTREM2-Fc with anti-IgG Fc agarose beads. These data show that the TREM2 ectodomain binds APOE and that there is no major difference in binding between the AD variants and WT TREM2 (Figs 1C and EV1C). In this experiment, a minor interaction between the human sIgG4-Fc domain and APOE3 is observed.

TREM2 is a member of a multiprotein family that is divergent between human and mice. We explored whether select members of

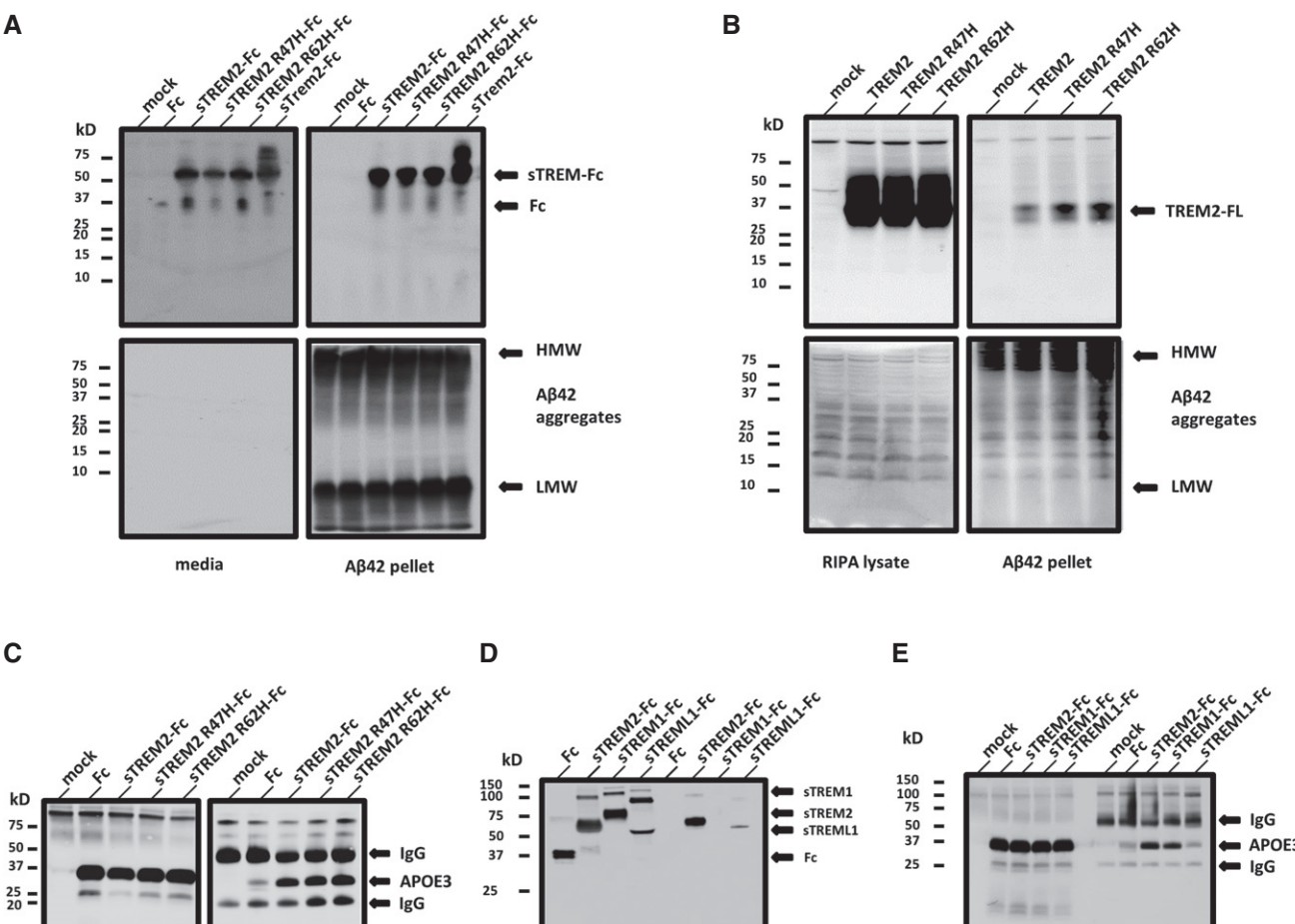

**Figure 1. Soluble TREM2 and soluble TREML1-Fc bind Aβ42 fibrils.**

A Media from soluble TREM2-Fc transfected HEK293T cells incubated with Aβ42 fibrils. Top left panel shows original media samples. Top right panel shows the presence of soluble TREM2 associated with Aβ42 pellet. The bottom two panels represent the same samples as the top, but probed with 6E10.

B RIPA lysate from TREM2 transfected HEK293T cells incubated with Aβ42 fibrils. Top left panel shows original RIPA lysate samples, and top right panel shows the presence of TREM2 associated with Aβ42 pellet. The bottom two panels represent the same samples as the top, but probed with 6E10.

C Media from soluble TREM2-Fc and APOE3 co-transfected HEK293T cells incubated with agarose beads anti-human IgG Fc. Left panels show original media samples. Right panels show the purified soluble TREM2-Fc (bottom) and the presence of APOE3 with the purified soluble TREM2-Fc (top).

D Media from soluble TREM-Fc family member transfected HEK293T cells incubated with Aβ42 fibrils. Top panel shows original media sample and the presence of TREM family members associated with Aβ42 pellet. Bottom panel shows original media and Aβ42 pellet probed with 6E10.

E Media from soluble TREM-Fc family members and APOE3 co-transfected HEK293T cells incubated with agarose beads anti-human IgG Fc. Top panel shows original media sample and the presence of soluble TREM family members associated with APOE3. Bottom panel shows original media and purified soluble TREM-Fc family members probed with anti-V5.

Data information: All experiments were replicated 3 independent times. Western blots in panels (A, B and C) have been cropped. The originals showing the supernatant fractions are available as Fig EV1.

Source data are available online for this figure.

the TREM2 family bind Aβ42 fibrils and APOE. We repeated the fAβ pull-down experiments in Fig 1A with two other soluble family members, which share low sequence homology with each and

other: TREM1-Fc and TREML1-Fc. In Fig 1D, we confirm presence of soluble TREM2-Fc and TREML1-Fc in the pelleted Aβ42 fibrils. Soluble TREM1-Fc was not detected in the pelleted fAβ. Soluble

TREM1-Fc like soluble TREM2-Fc interacts with APOE3, but there is lack of detectable interaction between soluble TREML1-Fc and APOE3 (Fig 1E).

### Soluble TREM2-Fc and AD variants interact with Aβ monomers or APOE with a similar affinity, and soluble TREM-Fc family members interact with Aβ42 oligomers with a higher affinity

We used BioLayer Interferometry (BLI) to further explore the interaction between Aβ and TREM2 and other TREM family members. By immobilizing the TREM proteins on the BLI sensor, ligand interaction can be monitored in real time, enabling precise determination of the association and dissociation constants. As poorly soluble fAβ does not produce detectable binding in this assay even with high-affinity control anti-Aβ antibodies, we were only able to assess binding with Aβ42 oligomers and Aβ40 and Aβ42 monomers. We initially explored interactions between Aβ42 oligomers and sTREM2-Fc in the BLI assay. These data revealed that sTREM2 bound Aβ42 oligomers with high affinity largely attributable to a very slow dissociation. In these initial studies, we also found no interaction between the Fc alone and Aβ42 oligomers. Further, when a high-affinity anti-Aβ1-16 antibody (Ab5) was incubated following the dissociation step, we were able to demonstrate that the Aβ oligomers were indeed tightly complexed with sTREM2 on the sensor (Fig EV2A). After further optimization of the assay (see methods), we systematically evaluated the binding of sTREM-Fc proteins to various concentrations of Aβ42 oligomers, Aβ42 monomers Aβ40 monomers, as well as lipidated recombinant APOE3 and APOE4 (Figs 2A and B, and EV2B). Under these optimized conditions, no sFc binding is observed, and non-specific binding of APOE3 or APOE4 eliminated. These studies enabled precise determination of the Kon, Kdis (Koff), and the overall KD (Fig 2C) of various form of Aβ and APOE under identical conditions. These studies revealed very high-affinity binding between Aβ42 oligomers and sTREM2-Fc and the AD variants (<1.0 E-12 M). This strong binding is largely attributable to the apparent irreversibility of the binding. Indeed, the dissociation constants are the maximal measurable (Kdis = 1.0 E-07 1/s) on the Octet Red instrument. In contrast to the binding observed between sTREM2 and Aβ42 oligomers, binding to Aβ42 monomers was much weaker and reversible (KD from 1.4 to 2.9 E-07 M). The KDs of the WT TREM2 were not statistically different from the AD variants. Binding to Aβ40 monomers was similar to binding to Aβ42 monomers (KDs from 1.4 to 2.7 E-07 M), and again, the KDs of the WT TREM2 were not statistically different from the AD variants. We also established KDs in the nm range for binding of lipidated APOE3 and APOE4 with sTREM2. KDs for APOE3 binding sTREM2 or the AD variants were very similar and not statistically different. Binding of APOE4 revealed a slight increase in binding affinity of R47H variant (1.7 fold) and the R62H variant (1.56-fold) with APOE4 that was statistically significant. Given the irreversible nature of the Aβ42 oligomers binding to sTREM2, we explored whether the binding of Aβ42 oligomers could block interaction with APOE3. We evaluated whether pre-incubation with Aβ42 oligomers to sTREM2 could block subsequent interaction with APOE3. As shown in Fig 3, binding of Aβ42 oligomers blocked APOE3 binding. In contrast, addition of Aβ42 oligomers to sTREM2 previously bound to APOE3 did not block subsequent

interaction with Aβ42 oligomers. We next examined the binding of select soluble mouse and human TREM family members with Aβ42 oligomers by BLI (Fig 2D). TREML1 showed a similar high-affinity binding of Aβ42 oligomers to TREM2 but the interaction between TREM1 and Aβ42 oligomers was much weaker (1.1E-7 M). In contrast, both mouse Trem1 and Treml1 bound Aβ42 oligomers with high affinity.

We further investigated the binding of fAβ with soluble TREM2-Fc and the AD variants by an ELISA binding assay due to their unsuitability with BioLayer Interferometry. We compared the binding of fAβ, Aβ42 monomers, and Aβ42 oligomers with soluble TREM2-Fc and the AD variants. In Fig EV3, Aβ42 fibrils incubated with soluble TREM2-Fc and the AD variants or soluble Trem2-Fc increased weakly the signal compared to the Fc control and showed no significant differences. The Aβ42 oligomers incubated with soluble TREM2-Fc or the AD variants or soluble Trem2-FC produced a signal significantly higher compared to Fc control and around two times stronger than the one from the Aβ42 monomers. In Fig 4A, we tested Aβ42 monomers at higher concentrations. A significant signal between Fc control and the soluble TREM2-Fc or the AD variants or soluble Trem2-Fc is obtained at 1,000 nM Aβ42 monomers and more. This result confirms the binding of Aβ42 monomers with soluble TREM2 and the AD variants and soluble Trem2-Fc without significant differences between all of them. We then repeated a similar ELISA assay with soluble TREM-Fc family members and Aβ42 oligomers. In Fig 4B, only soluble TREM2-Fc and soluble TREML1-Fc produced a significant signal compared to Fc control. This result confirms the binding of Aβ42 oligomers with soluble TREM2-Fc and soluble TREML1-Fc and lack of binding with soluble TREM1-Fc.

Collectively, these biochemical experiments confirm a direct interaction between Aβ42 and TREM2 and the AD variants TREM2 R47H or TREM2 R62H and Trem2 without any significant differences. Aβ42 oligomers have a stronger affinity for TREM2 than Aβ42 monomers. TREML1 also binds Aβ42 oligomers, but failed to bind APOE3 contrary to TREM2 or TREM1.

### Expression of TREM2 activates NFAT signaling and increases internalized Aβ42

We investigated whether the binding of Aβ42 oligomers with TREM2 activates cell signaling by exposing 2B4 NFAT-GFP reporter cells transduced with TREM2 to various concentrations of Aβ42 oligomers. As other TREM2 ligands activate NFAT in these cells, we predicted that the high-affinity interaction with Aβ42 oligomers would also induce NFAT signaling. Control cells exposed to Aβ42 oligomers weakly activated the GFP reporter compared to WT TREM2 expressing cells (Fig 5A). Cells expressing TREM2 R47H and exposed to Aβ42 oligomers also activated the GFP reporter, but at each concentration, the signal was lower than the signal in WT TREM2 transduced cells. Further study of reporter cells expressing WT TREM2, TREM2 R47H TREM2 R62H, or a putatively frontotemporal dementia (FTD) variant (TREM2 T96K) shows that the TREM2 AD variants R47H and R62H reduced NFAT activation whereas the FTD variants TREM2 T96K activated NFAT signaling at similar level then TREM2 (Fig 5B). These data demonstrate that TREM2 activates NFAT signaling in presence Aβ42 oligomers and this activation is reduced by the AD variants TREM2 R47H or R62H.

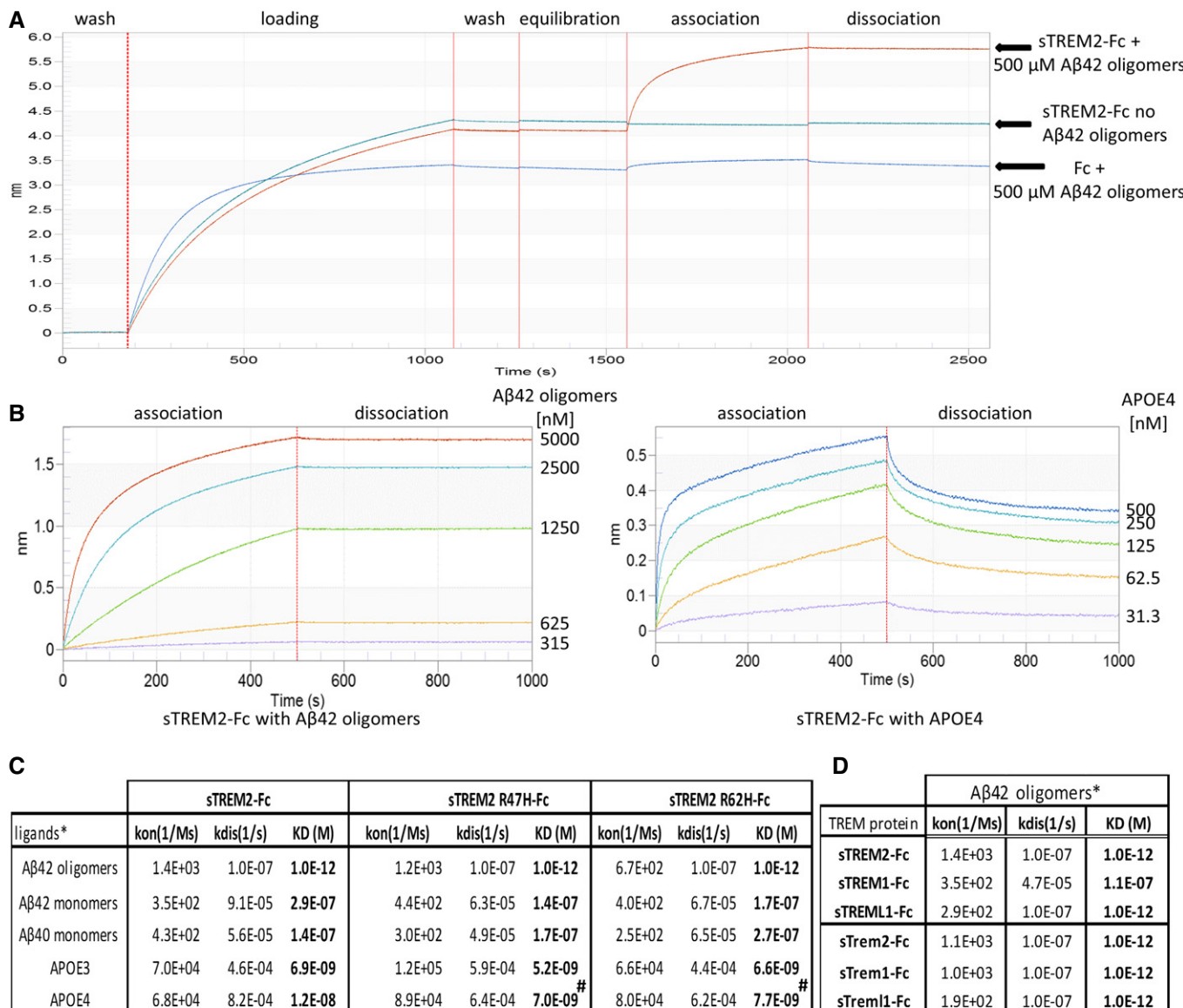

**Figure 2. Binding affinity of soluble TREM-Fc family members and AD variants with Aβ42 oligomers, Aβ42 monomers, Aβ40 monomers, and APOE3 and APOE4.**

BioLayer Interferometry experiments were performed to measure ligand binding affinity with soluble TREM-Fc family members and mouse soluble TREM-Fc family members.

A    Schematic representation of a full-length experiment.

B    Schematic representation of Aβ42 oligomers (right panel) and APOE4 (left panel) binding with soluble TREM2-Fc at various concentrations. Curves were fit by setting buffer control (no ligand) to $y = 0$.

C    Calculated Kon, Kdis, and KD values for Aβ42 oligomers or Aβ42 monomers, Aβ40 monomers, or APOE3 or APOE4 association/dissociation with soluble TREM2-Fc and AD variants. Kinetic constants were calculated using five different concentrations of ligand, and the entire experiment (including new ligand preparations and purifications of soluble TREM-Fc) was repeated twice ($^{\#}P < 0.05$, ANOVA, Tukey's multiple comparison test, see Appendix Table S1 for exact $P$-value).

D    Calculated Kon, Kdis, and KD value for Aβ42 oligomers association/dissociation with soluble human vs. mouse TREM-Fc family members. Kinetic constants were calculated using five different concentrations of ligand, and the entire experiment (including new ligand preparations and purifications of soluble TREM-Fc) was repeated twice.

Data information: Kon; constant of association, Kdis; constant of dissociation, KD; equilibrium constant of dissociation.

TREM2 expression in non-phagocytic cells confers phagocytic activity (Hsieh *et al*, 2009). To evaluate whether TREM2 expression induces Aβ42 uptake, we incubated Aβ42 over HEK293T cells transiently transfected with TREM2 and quantified internalized Aβ42. Compared to control transfection, TREM2 expression increased the amount of internalized Aβ42 (Fig 5C). TREM2 R47H and R62H also increased internalization levels relative to control, but the amount internalized was significantly less than that observed with WT TREM2 (Fig 5C).

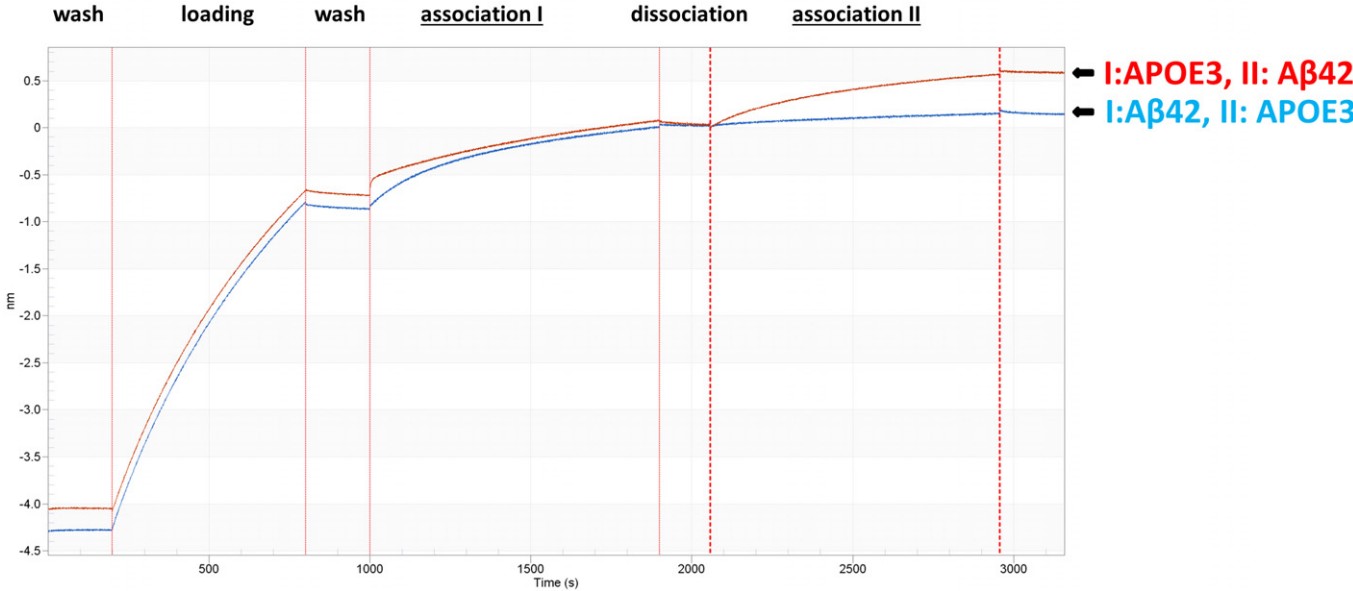

**Figure 3. Binding of Aβ42 oligomers on soluble TREM-Fc prevents binding of APOE3 in BioLayer Interferometry.**

BioLayer Interferometry experiments were performed to evaluate the binding capacity of APOE3 on soluble TREM2-Fc pre-incubated with Aβ42 oligomers. Schematic representations of two sequential associations of ligand on soluble TREM2-Fc are represented by I and II. Soluble TREM2-Fc loaded on a sensor is first incubated with APOE3 (association I) followed by a second incubation with Aβ42 oligomers (association II) or the reverse-incubated with Aβ42 oligomers (association I) followed by a second incubation with APOE3 (association II). All experiments were replicated 3 independent times.

We conducted several studies to further explore the reduced uptake observed by the AD variants. First, we investigated whether the shedding of TREM2 is affected by Aβ. If Aβ binding induces shedding of the TREM2 ectodomain and the AD variants are shed more, this could explain the decreased uptake observed. However, as observed in Fig 5D, despite similar levels of membrane-associated TREM2, TREM2 R47H, and TREM2 R62H, the amount of soluble TREM2 was similar and not altered by oligomeric Aβ. We also rigorously evaluated cell-surface levels of TREM2 using cell-surface biotinylation studies. These studies (Fig 5E) again revealed no significant differences in levels as previously observed (Kleinberger *et al*, 2014; Kober *et al*, 2016).

## Discussion

We find that TREM2 binds to various forms of Aβ with a very high-affinity interaction with Aβ42 oligomers. Further, TREM2 appears capable of mediating Aβ internalization and Aβ oligomers induce NFAT singling through TREM2. These data establish a direct link between Aβ aggregates, which are implicated as initiators of the pathological cascade in AD and TREM2 a microglial immune receptor. No difference in binding affinity between the AD-associated variants and WT TREM2 to any form of Aβ is detected. However, AD-associated TREM2 variants appear to reduce both Aβ internalization and downstream NFAT signaling. Consistent with other published data, these data indicate that TREM2 variants associated with AD risk may act through subtle loss of function (Song *et al*, 2018). These data are also consistent with the reported protection from AD attributable to a non-coding nucleotide polymorphism in

the TREM locus that increases both TREM2 and TREMl1 expression (Carrasquillo *et al*, 2017). Indeed, a recent modeling study showed that increased TREM2 levels reduced amyloid levels in an APP mouse model and attenuated a number of other amyloid-associated phenotypes in this model (Lee *et al*, 2018). As cell-surface levels of the AD variants and WT TREM2 are similar in these studies and there is no detectable difference in internalization or processing either in the presence or absence of Aβ, additional studies will be needed to better understand the functional differences between the AD variants and WT TREM2, which our data strongly suggest are downstream of the initial binding event.

The differential binding affinities between various forms of Aβ and TREM2 are interesting. Binding to monomeric Aβ is relatively weak, whereas binding to oligomers is essentially irreversible. Indeed, prior interaction of Aβ oligomers blocks binding of a second ligand, lipidated APOE. We have been unable to directly assess affinities to fAβ by BLI due to technical reasons, but do observe strong interaction in the fAβ pull-down assay in the absence of detergent. In the presence of detergent or by ELISA, which also contains small amount of detergent in the binding buffer, we observe a weaker interaction. Collectively these data support a model in which various forms of Aβ interact with TREM2 with different affinity and that difference in affinity appears to be attributable to conformational differences.

Given (i) the polarized localization of TREM2 on the microglial cells surrounding amyloid plaques (Jiang *et al*, 2014; Yuan *et al*, 2016; Jay *et al*, 2017) and (ii) the lack of homing of microglial to plaques that is observed in knockout mouse models (Ulrich *et al*, 2014; Jay *et al*, 2015, 2017; Wang *et al*, 2015, 2016; Yuan *et al*, 2016), it is intriguing to consider that high-affinity interaction

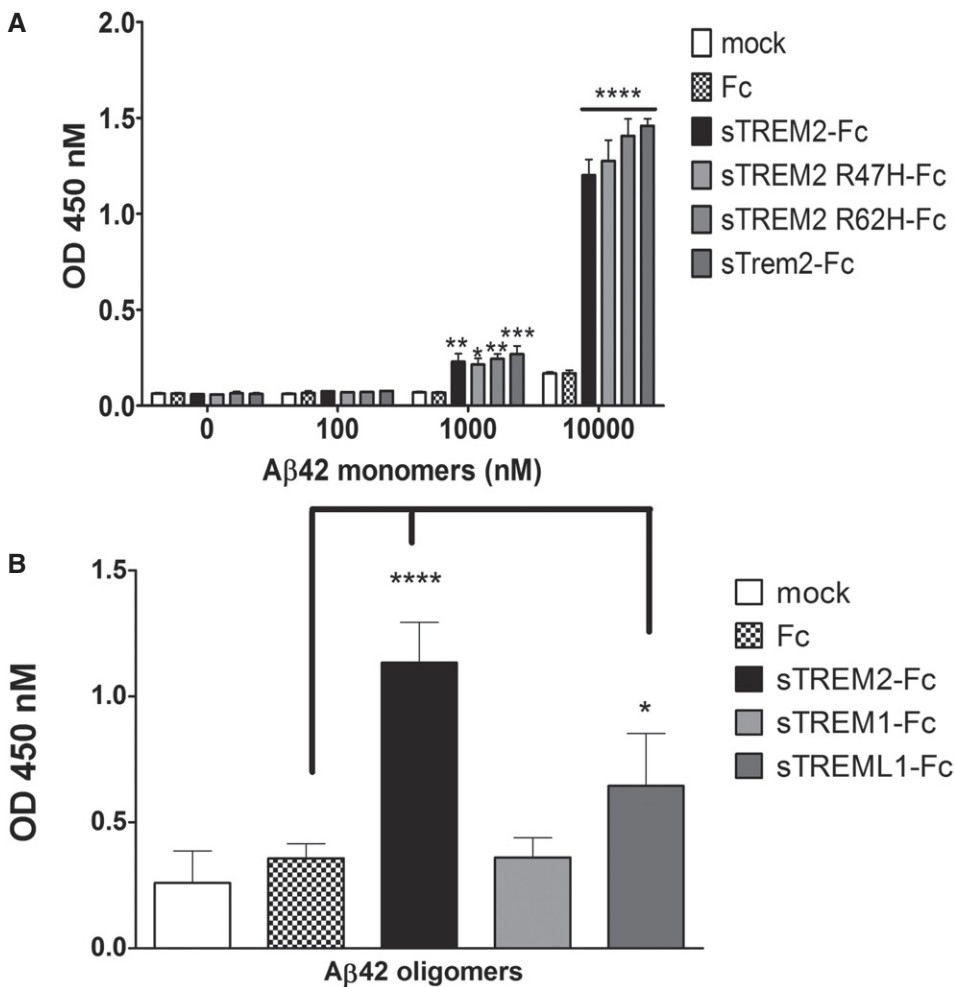

**Figure 4. Binding of Aβ42 monomers and Aβ42 oligomers with soluble TREM2-Fc measured by ELISA assay.**

ELISA plates were coated with purified soluble TREM2-Fc and then incubated with various concentrations of Aβ42 monomers or 1 μM Aβ42 oligomers.

A    Binding of Aβ42 monomers with soluble TREM2-Fc shows a dose-dependent signal. Results were expressed as the OD450 averaged ± standard error ($n = 3$; *$P < 0.05$, **$P < 0.01$, ***$P < 0.001$, ****$P < 0.0001$, two-way ANOVA, Bonferroni multiple comparisons, see Appendix Table S2 for exact $P$-values).

B    Binding of 1 μM Aβ42 oligomers with soluble TREM-Fc family members. Results were expressed as the OD450 averaged ± standard error ($n = 4$, *$P < 0.05$, ****$P < 0.0001$, ANOVA, Bonferroni multiple comparisons, see Appendix Table S2 for exact $P$-values).

between Aβ aggregates and TREM2 could underlie these observations. Although Aβ oligomers are described as soluble, immunohistochemical studies with select anti-oligomeric antibodies suggest that Aβ in an oligomeric conformation is present within plaques (Koffie *et al*, 2009). This raises an interesting speculation that TREM2 could interact with this plaque-associated oligomeric conformer of Aβ resulting in cell-surface polarization and microglial homing (Tanzi, 2015; Yeh *et al*, 2017). Given the promiscuous ligand specificity of TREM, and the apparent ability of Aβ oligomers to block subsequent ligand binding, it is hard to predict what the biological effects of engagement of TREM2 by Aβ oligomers would be in the peri-plaque region. Depending on relative concentration of ligands and relative efficiency of ligand-induced signaling the net result could be either prolonged TREM2 activation or reduced TREM2 activation.

Human TREM2 and mouse Trem2 are 68% identical and 76% homologous, and they bind Aβ oligomers with similar affinity.

These data would suggest that genetic manipulations of mouse Trem2 are likely to be informative with respect to the interactions with Aβ. Although TREM2 is the most abundant TREM family member based on expression levels in the brain, TREML1 is present at appreciable levels (Carrasquillo *et al*, 2017). Further, mouse Treml1 expression in the brain is dramatically increased by the transgene inserted to knockout TREM2 in one of the TREM2 mouse knockout models (Kang *et al*, 2018). Given that Treml1 shows strong interaction with Aβ oligomers, this could contribute to the discrepancies reported in the various knockout models with respect to effects on amyloid deposition. Our studies of human TREM1 and TREML1 show that TREML1 binds Aβ oligomers with high affinity but TREM1 does not. Conversely, TREM1 binds lipidated APOE but TREML1 does not. In contrast, both mouse Treml1 and Trem1 bind Aβ oligomers. These data can provide a basis for understanding both the larger biological role of TREM2 and family members in AD but also provide a

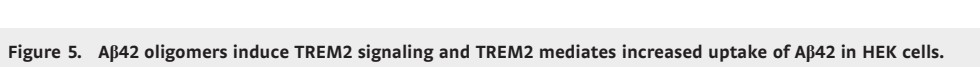

**Figure 5. Aβ42 oligomers induce TREM2 signaling and TREM2 mediates increased uptake of Aβ42 in HEK cells.**

A   2B4 NFAT-GFP reporter cells transduced with TREM2 or AD variant R47H were incubated with various concentrations of Aβ42 oligomers for 12 h. GFP expression was detected at 1 μM Aβ42 oligomers with TREM2 transduced cells. No GFP expression was detected with Aβ42 oligomers from control transduced cells. Results were expressed as the percentage GFP cells averaged ± standard error (n = 3, *P < 0.05 compared with mock, ****P < 0.0001 compared with mock, #P < 0.05 compared with TREM2, ####P < 0.0001 compared with TREM2, two-way ANOVA, Bonferroni multiple comparisons, see Appendix Table S3 for exact P-values).

B   B-2B4 NFAT-GFP reporter cells transduced with TREM2 or variants (R47H, R62H, and T96K) were incubated with 10 μM Aβ42 oligomers for 12 h. No GFP expression was detected with Aβ42 oligomers from control transduced cells. Results were expressed as the percentage GFP cells averaged ± standard error (n = 3, *P < 0.05 compared with TREM2, **P < 0.05 compared with TREM2, ****P < 0.0001 compared with TREM2, ##P < 0.01 compared with TREM2 T96K, ###P < 0.001 compared with TREM2 T96K, ####P < 0.0001 compared with TREM2 T96K, ANOVA, Dunnett's multiple comparisons test, see Appendix Table S3 for exact P-values.

C   HEK293T cells transfected with TREM2 or AD variant incubated with 500 nM Aβ42 monomers were lysed for Aβ42 quantifications by ELISA. Expression of TREM2 increases Aβ42 level detected compared to mock control as well the AD variant. Results were averaged ± standard error (n = 9, #P < 0.05 compared with TREM2, ##P < 0.01 compared with TREM2, ***P < 0.001 compared with mock, ****P < 0.0001 compared with mock, ANOVA, Tukey's multiple comparison test, see Appendix Table S3 for exact P-values).

D   HEK293T cells transfected with TREM2 were incubated with Aβ42 oligomers for 16 h. Conditioned media and cell lysate were analyzed by Western blot for TREM2 (RIPA lysate) and soluble TREM2 (media) detection. Experiment was replicated 3 independent times.

E   HEK293T cells transfected with TREM2 were subjected to cell membrane biotinylation for purification. Bottom panel in (D) shows the presence of TREM2 in the purified product. Endogenous APP and β-tubulin were used as controls for the purification (middle and upper panels). Experiment was replicated 3 independent times.

Source data are available online for this figure.

framework to understand the structural features of the various TREM molecules required for Aβ interaction.

We confirmed the previously reported binding between TREM2 and APOE, although some of the details that we report regarding binding are distinct (Atagi *et al*, 2015; Bailey *et al*, 2015; Yeh *et al*, 2016). Whereas other groups have reported reduced binding of APOE with TREM2 AD variants (Atagi *et al*, 2015; Bailey *et al*, 2015; Yeh *et al*, 2016), we find little difference between the binding affinity between lipidated APOE and TREM2 or the AD variants. In fact, for APOE4 binding, we see slightly increased in affinity for the AD variants compared to WT TREM2. Though statistically significant, this is relatively small difference and unlikely to translate into biological differences with respect to function. Differences between our data and previous reports may be due to technical differences. Yeh *et al* (2016) make claims about binding affinity based on BLI but did not actually measure association and dissociation constants. Yuka *et al* evaluated binding of TREM2/APOE by dot blot and ELISA assays, which again do not provide precise data on kinetics of binding, and Bailey *et al* reported that TREM2 R47H reduced the affinity of TREM2 for APOE used ELISA assay (Atagi *et al*, 2015; Bailey *et al*, 2015). In contrast to Yuka *et al*, who reported a lack of interaction between TREM1 and APOE in dot blot assay, we detect an interaction between secreted TREM1 and APOE3 from media. (Atagi *et al*, 2015).

An interaction between TREM2 and Aβ42 oligomers has also recently been reported by several other groups (Zhao *et al*, 2018; Zhong, 2018). Our data and these reports consistently show stronger binding between TREM2 and Aβ oligomers compared to monomers (Zhao *et al*, 2018). In contrast to these other studies, we do find reproducible and appreciable binding between monomeric Aβ and TREM2, though the interaction is weaker than the interaction with a known ligand APOE and much weaker than the interaction with oligomeric Aβ. We also observe a much higher binding affinity between oligomeric Aβ and TREM2 than what is reported in these other studies. In our studies, this high affinity is primarily driven by a Kdis that is the lowest measurable on the Octet instrument. A prediction of the extremely low Kdis is that the nearly irreversible binding of oligomeric Aβ to TREM2 would block interaction with another ligand. We tested this prediction with APOE and find that this is indeed the case—prior incubation of Aβ oligomers with TREM2 blocked subsequent binding of APOE to TREM2. A final and important discrepancy between these reports is that we reproducibly do not detect differences in affinity between various forms of Aβ and TREM2 WT and the AD variants in either the BLI studies or in ELISA assays. Notably, although these other reports use BLI to calculate affinity of TREM2 WT to oligomeric Aβ, only solid-phase plate binding assays are used to show differences in the affinity between Aβ42 oligomers and TREM2 or the TREM2 variants (Zhao *et al*, 2018; Zhong *et al*, 2018). Clearly, many factors ranging from binding buffers to protein purity and also form Aβ oligomers can alter affinity measurements. It is likely that such differences may account for the quantitative differences in binding affinity observed in our study compare to these other studies. Additional studies will be needed to reconcile these differences, which may have important pathophysiologic implications. Indeed, we show consistent effects of AD variants on both Aβ internalization and

NFAT signaling both consistent with a partial loss of function. However, our data indicate that these functional effects are not attributable to decreased affinity for Aβ under the conditions tested. The functionality of interaction between Aβ oligomers and TREM2 showed an activation of NFAT signaling as observed by Zhong *et al* (2018). This signaling activation was indirectly confirmed by showing that Aβ42 oligomers enhanced the interaction between TREM2 and DAP12 and induced SYK phosphorylation, an important factor for NFAT signaling (Zhao *et al*, 2018).

Reports exploring TREM2 deficiency in mouse models of Aβ deposition have shown both differing effects on Aβ levels during early and late stages and in some cases no change in overall amyloid loads (Ulrich *et al*, 2014; Jay *et al*, 2015, 2017; Wang *et al*, 2015, 2016). Studies that showed effects on Aβ deposition reported changes in Aβ loads were rather modest and regional in nature (Ulrich *et al*, 2014; Wang *et al*, 2016). This suggests that any interaction between TREM2 and Aβ (or perhaps even co-deposited APOE) in the peri-plaque region alters microglial response to the plaque with a minimal effect on plaque clearance (Jiang *et al*, 2014; Jay *et al*, 2015, 2017; Wang *et al*, 2015; Yuan *et al*, 2016). Given this modeling data, we would suggest that any interaction between TREM2 and Aβ (or perhaps even co-deposited APOE) in the peri-plaque region alters microglial response to the plaque but would have minimal effect on plaque clearance. Notably, recent data from one of our group show that R47H TREM2 variant fails to restore microglial homing and activation around amyloid plaques in the mouse knockout background (Song *et al*, 2018). They also found that sTREM2-R47H despite being present at equal levels as a common variant (CV) of sTREM2 was not associated with plaques or neurons, whereas the sTREM2-CV was. Another group showed that AD mice Trem2 R47H heterozygous reduced myeloid cell responses to amyloid deposition and reduction in proliferation and CD45 expression around plaque (Cheng-Hathaway *et al*, 2018).These data present a conundrum given that we showed kinetics of binding of the R47H and WT TREM2 to Aβ are indistinguishable. Further, they are hard to reconcile functionally with the increased shedding of the Han Chinese TREM2 variant that is reported to increase ectodomain shedding, but would not be expected to alter the functionality of the ectodomain (Schlepckow *et al*, 2017; Thornton *et al*, 2017). The overexpression of TREM2 appears protective. Previous studies showed protective functions of TREM2 overexpression by facilitating Aβ42 phagocytosis and inhibited proinflammatory responses in cultured primary microglia, reducing AD-related neuropathology in mice APPswe/PS1dE9 and ameliorating tau pathology a tau mice model (Jiang *et al*, 2014, 2016b). Our results tend in the same direction by showing benefits of TREM2 expression like increasing Aβ42 phagocytosis and activating cell signaling.

We present biochemical and cellular data that demonstrate that TREM2 and select family members can bind to various forms of Aβ with the highest affinity interaction observed for Aβ oligomers, and we confirm an interaction between TREM2 and APOE (Atagi *et al*, 2015; Bailey *et al*, 2015; Yeh *et al*, 2016). Intracellular signaling elicited following binding by Aβ oligomers is attenuated by AD-associated TREM2 variants, though the AD variants appear to bind both Aβ and lipidated APOE with similar affinity to interaction with WT TREM2. These data link TREM2 to physical interactions with two

key molecules in Alzheimer's disease and provide clues that can illuminate the clearly complex biology of TREM2 in health and disease.

# Materials and Methods

HEK293T cells were grown in DMEM supplemented with Hyclone 10% fetal bovine serum (GE, Utah, USA) and 1% penicillin/streptomycin (Life Technologies, NY, USA). Cells were transiently transfected with calcium phosphate. 2B4 reporter cells were maintained in RPMI (Sigma, MO, USA) supplemented with 10% FBS, 1% GlutaMAX (Gibco, NY, USA), 1 mM sodium pyruvate (Corning, VA. USA), 1% penicillin/streptomycin (Gibco, NY, USA), and transfected by retroviral transduction.

### Antibodies

Antibodies are listed in Table 1.

### cDNA constructions

pcDNA5-FRT-TO-TREM2, pcDNA5-FRT-TO-TREM2 R47H, pcDNA5-FRT-TO-TREM2 R62H were a gift from Christian Haass (Kleinberger *et al*, 2014). Soluble domains of human TREM2, TREM1, TREML1 and mouse Trem2, Trem1, and Treml1 were cloned in frame with human IgG Fc4 and V5 epitope in pTR2-CBA plasmid (GeneScript, NJ, USA). TREMs cDNA constructions are listed in Table 2.

### Aβ42 fibrils pull down

Aβ42 fibril preparation and pull-down assay were performed as described (Chakrabarty *et al*, 2015; Pinotsi *et al*, 2016). Conditioned media with 5 mM EDTA or cellular lysate (RIPA SDS containing buffer) were centrifuged at 18,000 *g* for 5 min at 4°C to remove insoluble materials. Supernatants were incubated with Aβ42 fibrils for 1 h at room temperature. Aβ42 fibrils were centrifuged at 18,000 *g* for 5 min at 4°C to pellet the fibrils. Fibrils were washed with RIPA buffer or PBS (for conditioned media samples) and centrifuged again. The pelleted Aβ42 fibrils were then dissolved in protein loading buffer (31.25 mM Tris–HCl pH 7.5, 2% LDS, 10% glycerol, 1.5% β-mercaptoethanol, and Orange G). Cell conditioned media samples or cell lysates were also incubated with the fibril assay buffer to verify proteins aggregations. Dissolved fibrils were heated at 95°C for 5 min and loaded on Bis–Tris precast gels (Bio-Rad, CA, USA) and transferred on PVDF membrane for Western blotting. TREM2 was detected with HA antibody, soluble trems-Fc-V5 were detected with V5 antibody, and fibrils were detected with 6E10 antibody.

### Western blotting

PVDF membranes were blocked in TBS 0.5% casein a 1 h at room temperature. Primary antibody diluted in TBS with 0.2% Tween-20 (TBS-T) was incubated on the membrane 1 h at room temperature. The membranes were washed three times 5 min with TBS-T. Secondary antibodies were diluted in TBS-T and incubated 1 h at room temperature. The membranes were washed 3 times 5 min

**Table 1. List of antibodies used for Western blot and ELISA.**

| Antibodies | Source | Usage, dilution |
|---|---|---|
| TREM2 CTF | Cell Signaling | Western blot, 1/2,500 |
| HA tag | Cell Signaling | Western blot, 1/2,500 |
| V5 tag | Invitrogen | Western blot, 1/5,000 |
| Ab947 (APOE) | Chemicon | Western blot, 1/2,500 |
| HRP-Anti-Human IgG Fcγ | Jackson ImmunoResearch Inc | ELISA (for detection), 1/2,000 |
| 2.1.3 (Aβ42) | Todd. E. Golde | ELISA (for coating), 50 µg/ml |
| HRP-4G8 | BioLegend | ELISA (for detection), 1/2,000 |
| IRDye 800CW | LI-COR | Western blot (for detection), 1/12,500 |
| Alexa Fluor 680 goat or rabbit | Invitrogen | Western blot (for detection), 1/25,000 |
| 6E10 (amyloid) | BioLegend | Western blot, 1/1,000 |

with TBS-T and then analyzed with Odyssey infrared imaging system (LI-COR Inc., NE, USA).

### Purification of secreted soluble TREM-Fc-V5 family members from cell conditioned media

Conditioned media from HEK293T-transfected cells was centrifuged 5 min at 800 *g* to clear cells and then transferred to a tube with 0.5% Triton X-100, complete protease inhibitors, and 5 mM EDTA. Samples were incubated 2 h at room temperature with anti-human IgG Fc-specific agarose beads (Sigma, MO, USA) and washed three times with PBS. Purified proteins from agarose beads were washed in PBS and eluted with 100 mM glycine pH 2.7 followed a neutralization (pH~7.5) with 1 M Tris–HCl pH 8.8. Alternatively, agarose beads were pre-blocked with 5% bovine serum albumin (Sigma, MO, USA) 2 h at room temperature and washed with PBS before the purification. Purified samples were verified by Western blot to confirm protein purification.

### APOE interaction assay

HEK293T cells were co-transfected with soluble TREM2-Fc family members and APOE3. The soluble TREM-Fc family members were purified as described above with BSA pre-blocked agarose beads anti-human IgG Fc specific. Agarose beads were washed with PBS 0.5% Triton and eluted with protein loading buffer. Eluates were loaded on Bis–Tris precast gels and transferred on PVDF membrane for Western blotting. Soluble TREM2 Fc-V5 and AD variants were detected with V5 antibody, and APOE3 was detected with Ab947 antibody.

### BioLayer interferometry

Soluble TREM2-Fc-V5 and TREM2-Fc-V5 family proteins were purified as described above with anti-human IgG Fc-specific agarose beads. Aβ42 monomers and oligomers were prepared as described (Stine *et al*, 2003). Kinetic determination of TREM2/Aβ42 oligomer was initially performed on the Octet RED384 instrument (ForteBio

**Table 2. List of TREMs cDNA constructs used for the experiments.**

| Plasmid | Species | Notes | Epitope | Usage |
|---|---|---|---|---|
| sTREM2-Fc | Human | Soluble ectodomain | V5 | Figs 1A, C, D, and E, 2, 3, and 4 |
| sTREM2-R47H-Fc | Human | Soluble ectodomain AD variant | V5 | Figs 1A and C, 2C, and 4 |
| sTREM2-R62H-Fc | Human | Soluble ectodomain AD variant | V5 | Figs 1A and C, 2C, and 4 |
| sTREM1-Fc | Human | Soluble ectodomain | V5 | Figs 1D and E, 2D, and 4B |
| sTREML1-Fc | Human | Soluble ectodomain | V5 | Figs 1D and E, 2D, and 4B |
| sTrem2-Fc | Mouse | Soluble ectodomain | V5 | Figs 1A, 2D, and 4A |
| sTrem1-Fc | Mouse | Soluble ectodomain | V5 | Fig 2D |
| sTreml1-Fc | Mouse | Soluble ectodomain | V5 | Fig 2D |
| TREM2 | Human | Full-length protein | HA and FLAG | Figs 1B and 5C–E |
| TREM2-R47H | Human | Full-length protein AD variant | HA and FLAG | Figs 1B and 5C–E |
| TREM2-R62H | Human | Full-length protein AD variant | HA and FLAG | Figs 1B and 5C–E |
| TREM2-IRES-DAP12 TREM2-R47H-IRES-DAP12 | Human | Full-length protein | No epitope | Fig 5A and B |
| TREM2 R62H-IRES-DAP12 TREM2 T96K-IRES-DAP12 | Human | Full-length protein | No epitope | Fig 5B |

(Pall), CA, USA), using anti-human Fc-specific sensors (AHC, ForteBio (Pall), CA, USA), PBS 0.002%, and Tween-20 as assay buffer. Briefly, measurements were performed at 30°C and sensors were activated in a 96-well microplate in PBS and agitated at 1,000 rpm for 10 min. Various dilutions of Aβ42 or human lipidated APOE3 and APOE4 (gift from Dr. David M. Holtzman) were loaded in a 384-well microtiter plate.

A second BLI assay was developed for compatibility APOE proteins utilizing Protein A sensors (ProA, ForteBio (Pall). CA, USA). Protein A sensors were loaded to near-saturation with soluble TREM-Fc-V5 family proteins in PBS, transferred to fresh PBS for baseline measurement, then associated with Aβ42 oligomer, human lipidated APOE3 or APOE4 ligands along a serial dilution. The sensors were finally moved back to PBS for disassociation. Constant of association (kon), constant of dissociation (kdis), and equilibrium constant of dissociation (KD) values were determined by global fitting of the binding curves for each ligands dilutions and calculated by applying a 1:1 interaction model (fitting local, full) using the ForteBio Data Analysis software (CA, USA) version 9.0.0.14.

## ELISA binding assay

Soluble TREM-Fc family member proteins were purified and eluted as previously described for the BioLayer Interferometry. Purified samples were coated overnight at 4°C in 96-well plates (Immulon 4HBX, Thermo Scientific, NY, USA) in 100 mM NaCO$_3$ pH 11. Plates washed with PBS twice and then blocked 3 h at room temperature with blocking buffer (Block Ace, Abd Serotec, Japan). Blocked samples were incubated with Aβ42 monomers, Aβ42 oligomers, or Aβ42 fibrils at specified concentration for 1 h at room temperature. Aβ was detected with 4G8-HRP, and soluble TREM-Fc samples were detected with anti-human IgG HRP to insure similar coating level of proteins. Samples were developed with TMB substrate (KPL, Maryland, USA), and the reaction was stopped by adding 6.67% O-phosphoric acid (85%). The signal produced by HRP and TMB substrate was quantified by measuring the absorbance at 450 nM.

## Aβ cellular uptake

HEK293T transfected cells from a 6-well plate were incubated 3 h with 500 nM Aβ42 monomers. Cells were washed two times with PBS, incubated with trypsin 5 min, and centrifuged 3 min at 500 g. Supernatant was removed, and cells were washed three times with a large volume of PBS (10–15 ml/wash). Pelleted cells were lysed with 100 μl SDS 2% for 5 min, and then, 200 μl of PBS was added. Samples were sonicated and centrifuged at 18,000 g for 15 min at 4°C. Samples were diluted 1/40 in EC buffer (PBS containing 0.1 mM EDTA, 1% BSA, 0.05% CHAPS, pH 7.4, and 0.2% sodium azide) and analyzed by Sandwich ELISAs for Aβ42 as previously described (Ran et al, 2014).

## Biotinylation

Biotinylation experiments were performed to detect cell-surface expression of TREM2 and the AD variants. Briefly, transfected HEK293T cells were incubated with 2 mg/ml sulfo-NHS-biotin (Pierce) in PBS for 30 min on ice. Cells were washed with PBS 100 mM glycine, washed twice with PBS and then lysed in RIPA buffer. Cell lysates were incubated with streptavidin agarose resin (Pierce) overnight at 4°C and washed five times with RIPA buffer and then eluted with protein loading buffer. Eluates were analyzed by Western blot.

## NFAT activation assay

Nuclear factor of activated T-cell (NFAT) activation assay was performed with a GFP reporter as described previously (Wang et al, 2015). The 2B4 reporter cell lines transfected with TREM2-IRES-DAP12 or TREM R47H-IRES-DAP12 were incubated with Aβ42 oligomers for 12 h. NFAT signaling activation was measured by the expression of GFP which was analyzed by flow cytometry. Reporter activity is presented as percentage of cells that express GFP.

## Statistical analysis

Statistics were performed with GraphPad Prims (CA, USA) version 7. Details on specific tests used are providing in the Figure legends.

**Expanded View** for this article is available online.

### The paper explained

**Problem**

Triggering receptor expressed on myeloid cells 2 (TREM2) is a member of the immunoglobulin superfamily highly expressed in microglial cells. Rare coding variants of TREM2 (R47H, R62H) are associated with increased risk for Alzheimer's disease (AD), but how they confer this risk remains uncertain.

**Results**

We assessed binding of TREM2, AD-associated TREM2 variants and select TREM family members to various forms of Aβ and APOE in multiple assays. TREM2 interacts directly with various forms of Aβ in multiple assays, with highest affinity interactions observed between TREM2 and soluble Aβ42 oligomers. AD-associated TREM2 variants bind various forms of Aβ with similar affinity as WT TREM2. High-affinity binding of TREM2 to Aβ oligomers (KD < 1e-12) is characterized by very slow dissociation. Pre-incubation with Aβ oligomers is shown to block the lower affinity interaction of APOE. Our data show that mouse Trem2, Trem1 and Treml1 show the same high-affinity interaction with Aβ oligomers as human TREM2. In cellular assays, AD-associated variants of TREM2 reduced the amount of Aβ42 internalized, and in NFAT reporter assay, the R47H and R62H variants decreased NFAT signaling activity in response to Aβ42.

**Impact**

These studies identify a high-affinity interaction between TREM2 and Aβ that can block interaction with another ligand, APOE. These data link TREM2 to physical interactions with two key molecules in Alzheimer's disease, Aβ and APOE, and suggest that partial loss of function of TREM2 mediates increased risk for AD.

## Acknowledgements

This work was supported by NIH grants RO1AG32991 and NIA grant AG046139-01. We thank Dr. Christian Haass for providing the TREM2 plasmid.

## Author contributions

CBL, SLM, YZ, TBL, PEC, YR, and TEM performed experiments. CBL, TBL, PC, DMH, JDU, MC, and TEG designed the experiments, reviewed the manuscript, and edited the manuscript drafts. CBL and TEG wrote the manuscript.

## Conflict of interest

The authors declare that they have no conflict of interest.

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
