## [Review Process File · EMBO Molecular Medicine]

High affinity interactions and signal transduction between A β oligomers and TREM2

Christian B. Lessard, Samuel L. Malnik, Yingyue Zhou, Thomas B. Ladd, Pedro E. Cruz, Yong Ran, Thomas E. Mahan, Paramita Chakrabaty, David M. Holtzman, Jason D Ulrich, Marco Colonna, Todd E. Golde

Review timeline:

Submission date:	21 February 2018
Editorial Decision:	14 May 2018
Revision received:	20 July 2018
Editorial Decision:	04 September 2018
Revision received:	13 September 2018
Accepted:	17 September 2018

Editor: Céline Carret

Transaction Report:

1st Editorial Decision

14 May 2018

Thank you for the submission of your manuscript to EMBO Molecular Medicine, and most of all thank you for your continued patience. We have now heard back from the two referees whom we asked to evaluate your manuscript.

You will see from their attached comments that the referees find the topic timely and important. However, they also highlight a number of issues and limitations that must be addressed, experimentally when needed. While both referees mention the papers from Zhao et al and Zhong et al, both published in March 2018, we would like to encourage you to critically discuss these papers as recommended by referee 2, but nothing further-reaching will be asked as you are under our scooping protection policy.

Please note that EMBO Molecular Medicine strongly supports a single round of revision and that, as acceptance or rejection of the manuscript will depend on another round of review, your responses should be as complete as possible.

I look forward to receiving your revised manuscript.

***** Reviewer's comments *****

Referee #1 (Comments on Novelty/Model System for Author):

No major advance in the field. Technical flaws.

Referee #1 (Remarks for Author):

Lessard et al. characterize the binding of A β to human wildtype TREM2, the AD associated variants R47H and R62H as well as mouse Trem2. By using a BioLayer Interferometry assay they show equal KD values for binding of A β 42 oligomers to all soluble TREM2/Trem2 variants. Analysis of A β 40 and 42 monomers revealed also equal KD values for binding to all soluble TREM2 variants but showed weaker binding as compared to oligomers. Binding of fibrillar A β was analyzed by ELISA and pulldown assays and showed no difference between the TREM2 variants. Finally, the authors treated TREM2-transfected NFAT-reporter cells with oligomeric A β and detected a decreased signaling via the TREM2 R47H variant as compared to wildtype TREM2 as well as a reduced A β uptake.

While the calculation of KD values for monomeric and oligomeric A β is interesting, the paper presents with several weaknesses that should be addressed.

Major points:

1. The KD values presented in Fig. 2 do not show a difference for the binding of oligomeric A β 42 to all human TREM2 variants or mouse Trem2. However, Suppl. Fig. 3B shows a significant decrease in binding to mouse Trem2 by ELISA. This discrepancy should be explained.
2. To confirm specificity of the interaction of fibrillar A β 42 and TREM2/Trem2 a negative control with equal protein levels in the media should be included. Fig. 1B is missing a negative control as well.
3. In Fig. 3, the authors show that oligomeric A β 42 can block the subsequent binding of APOE3 to TREM2 while oligomeric A β 42 can still bind after pre-incubation of TREM2 with APOE3. This is an interesting finding and should be further expanded. How does the association constant of A β 42 change with and without APOE3 pre-incubation? Fig. 2C shows a small but significant difference in binding of APOE4 to the TREM2 variants. Does this influence the subsequent binding of oligomeric A β 42 in the BioLayer Interferometry assay?
4. Signaling via TREM2 after treatment with A β 42 oligomers was analyzed by means of a NFAT reporter cell line (Fig. 5). A significant decrease was shown for the R47H variant, but the R62H variant was not analyzed and should be included here.
5. Two recent publications already characterized the binding of oligomeric A β 42 to TREM2 variants in detail (Zhong et al., March 2018, Mol Neurodegener. and Zhao et al., March 2018, Neuron). Zhong et al. as well as Zhao et al. showed that oligomeric A β 42 is a ligand for TREM2 and that it shows a reduced binding to the R74H and R62H variants as compared to WT TREM2. This finding in these two independent publications is in sharp contrast to the results presented by Lessard et al. and should be discussed.
6. The two above mentioned publications report also some additional findings, which take away the novelty of the research presented in the study performed by Lessard et al. Zhong et al. showed that oligomeric but not monomeric A β 42 was able to activate NFAT signaling via TREM2. Furthermore, Zhao et al. report a reduced affinity of monomeric A β 42 - as compared to oligomeric A β 42 - to TREM2 and show that knockout of TREM2 does not affect A β uptake but degradation. Therefore, the authors should expand the scope of their study to make it more suitable for publication in a journal like EMBO Mol Med.. For example, the relationship with APOE variants could be explored in more detail or the mechanism of the differential downstream signaling and A β uptake (shown in Fig. 5) could be investigated.

Minor points:

1. Suppl. Fig. 1 is not mentioned anywhere in the text.
2. Molecular weight marker is missing in Suppl. Fig. 2B.

3. Figure 4 and 5 are not correctly referenced in the Results section.

Referee #2 (Comments on Novelty/Model System for Author):

In this manuscript, the authors describe experiments demonstrating that the interaction between TREM2 and A β oligomers can block interaction with another ligand and that AD-associated TREM2 variants bind A β with equivalent affinity but show loss of function in terms of signaling and A β internalization.

...

While the paper is timely and important this reviewer has several reservations regarding the data and their presentation.

1. It is difficult to evaluate the impact of the findings presented. While the binding experiments are convincing, they do not reveal a ground-breaking idea, as Abeta binding by TREM2 has been proposed by others as well (most recently, by Zhao et al., *Neuron* 2018; Zhong et al., *Mol Neurodegen* 2018). The concept that TREM2 variants affect the activation of downstream cascades, is interesting, is based on in vitro experiments alone, and is apparently inconsistent with the study of Kober et al. *J Mol Biol* 2016, showing that in R47H knock-in mice, TREM2 binding to Abeta oligomers is reduced. Similarly, the authors cite their own study (Song et al. *J Exp Med* 2018) in the Discussion as contradicting their present data, defining their own data as "a conundrum", but without attempting to provide clarity by explaining these contradictory results.

2. It is unclear how to interpret the NFAT assay. The authors should discuss this issue. Perhaps the author can address the effects of the purportedly protective T69K variant, as comparative NFAT assays, which may lend a stronger support to the partial loss-of-function hypothesis.

3. In general, the paper is not carefully written. Besides recurrent typos (and occasional poorly-constructed sentences), there are obvious portions of text in the Results section that would fit better in either the Methods (as in the first paragraph) or Discussion (as at the end of the final paragraph) sections.

1st Revision - authors' response

20 July 2018

Referee #1 (Comments on Novelty/Model System for Author):

No major advance in the field. Technical flaws.

Referee #1 (Remarks for Author):

Lessard et al. characterize the binding of A β to human wildtype TREM2, the AD associated variants R47H and R62H as well as mouse Trem2. By using a BioLayer Interferometry assay they show equal KD values for binding of A β 42 oligomers to all soluble TREM2/Trem2 variants. Analysis of A β 40 and 42 monomers revealed also equal KD values for binding to all soluble TREM2 variants but showed weaker binding as compared to oligomers. Binding of fibrillar A β was analyzed by ELISA and pull-down assays and showed no difference between the TREM2 variants. Finally, the authors treated TREM2-transfected NFAT-reporter cells with oligomeric A β and detected a decreased signaling via the TREM2 R47H variant as compared to wildtype TREM2 as well as a reduced A β uptake.

While the calculation of KD values for monomeric and oligomeric A β is interesting, the paper presents with several weaknesses that should be addressed.

Major points:

1. The KD values presented in Fig. 2 do not show a difference for the binding of oligomeric A β 42 to all human TREM2 variants or mouse Trem2. However, Suppl. Fig. 3B shows a significant decrease in binding to mouse Trem2 by ELISA. This discrepancy should be explained.

Answer

In supplement 3b, the statistical analysis showed no significant differences between sTREM2-Fc and strem2-Fc (multiple comparisons). To clarify, we mention in the legend “compared with Fc.”

2. To confirm specificity of the interaction of fibrillar A β 42 and TREM2/Trem2 a negative control with equal protein levels in the media should be included. Fig. 1B is missing a negative control as well.

Answer

Figure 1A has two controls. One control is the media from mock transfected cells. This control will reveal any nonspecific binding to the aggregates from the secreted proteins present in the media. The second control is the media from Fc transfected cells. This control will reveal if Fc proteins bind the aggregates. We did not use Fc fusion proteins for Figure 1B, but the full length TREM2 protein instead, from a cell lysate. The negative control was a cell lysate from mock transfected cells incubated with aggregates. Equal amounts of protein were loaded. We did not over expose our Western Blot and little differences in band intensity between the samples remain insignificant (WB remains semi-quantitative). Notably, we have previously used and published this pull down assay for validating specificity of anti-A β scFvs to A β fibrils (Levites et al J. Neuroscience 2006).

3. In Fig. 3, the authors show that oligomeric A β 42 can block the subsequent binding of APOE3 to TREM2 while oligomeric A β 42 can still bind after pre-incubation of TREM2 with APOE3. This is an interesting finding and should be further expanded. How does the association constant of A β 42 change with and without APOE3 pre-incubation? Fig. 2C shows a small but significant difference in binding of APOE4 to the TREM2 variants. Does this influence the subsequent binding of oligomeric A β 42 in the BioLayer Interferometry assay?

Answer

In vitro competition among sequentially associated A β oligomers and APOE3 for the TREM2 ligand is by nature a qualitative observation to confirm higher affinity of TREM2/42 oligomer. In a pre-incubation paradigm, we would expect a change in the on-rate given the time (however short) necessary for the APOE3 to be displaced. The change in KD may not be an indication of the completeness of the competition. We have not extensively evaluated this issue, but given the difference in TREM2 affinity for APOE is small, and the affinity of TREM2Fc for oligomeric A β under the same conditions is almost 3 orders of magnitude higher, we simply did not see the utility.

4. Signaling via TREM2 after treatment with A β 42 oligomers was analyzed by means of a NFAT reporter cell line (Fig. 5). A significant decrease was shown for the R47H variant, but the R62H variant was not analyzed and should be included here.

Answer

Similar studies with the variant R62H are now included. The mutants R62H, like R47H show a relative loss of downstream signaling following A β engagement.

5. Two recent publications already characterized the binding of oligomeric A β 42 to TREM2 variants in detail (Zhong et al., March 2018, Mol Neurodegener. and Zhao et al., March 2018, Neuron). Zhong et al. as well as Zhao et al. showed that oligomeric A β 42 is a ligand for TREM2 and that it shows a reduced binding to the R74H and R62H variants as compared to WT TREM2. This finding in these two independent publications is in sharp contrast to the results presented by Lessard et al. and should be discussed.

Answer

Our paper was already submitted when these reports were available online. We updated our discussion.

6. The two above mentioned publications report also some additional findings, which take away the novelty of the research presented in the study performed by Lessard et al. Zhong et al. showed that oligomeric but not monomeric A β 42 was able to activate NFAT signaling via TREM2. Furthermore,

Zhao et al. report a reduced affinity of monomeric A β 42 - as compared to oligomeric A β 42 - to TREM2 and show that knockout of TREM2 does not affect A β uptake but degradation. Therefore, the authors should expand the scope of their study to make it more suitable for publication in a journal like EMBO Mol Med.. For example, the relationship with APOE variants could be explored in more detail or the mechanism of the differential downstream signaling and A β uptake (shown in Fig. 5) could be investigated.

Answer

We agree with the reviewer about the novelty. However, our manuscript was submitted to EMBO Mol Med before the publication of these papers. EMBO Molecular Medicine has a "scooping protection" policy, whereby similar findings that are published by others during review or revision are not a criterion for rejection.

Zhong et al. showed that neither monomeric nor scrambled A β bound to TREM2 and Zhao et al. showed a binding of A β 1-42 monomers to TREM2. Zhong et al. and Zhao et al. provided contradicting results about the binding of A β 42 monomers on TREM2. Our results show that A β 1-42 monomers and oligomers bind TREM2, but the binding affinity of monomers is weaker than oligomers.

As mentioned by the reviewer here, "knockout of TREM2 does not affect A β uptake but degradation". Indeed, Zhao et al. showed no differences in FAM A β 1-42 uptake between WT and TREM2 KO microglia in their experimental conditions. Several reasons can explain this: they use a phagocytosis cell line and they use aggregated amyloid at 200 nM only. Aggregates are very sticky and it is possible to observe no differences between microglia WT of TREM2 -/- in the uptake amount under these conditions. We used A β monomers and measured the uptake after 3 hours. Our experimental conditions are different and we show clearly that TREM2 increases A β 42 uptake.

Our results are clear and we show binding of A β 42 between TREM2, and physiological roles of this binding: 1- increases A β 42 uptake by the cells, 2- activates cell signaling. We also show binding of the other TREM family members with amyloid which should be considered in the evaluation process.

Minor points:

1. Suppl. Fig. 1 is not mentioned anywhere in the text.

Answer

It is mentioned in the figure legend. We now have a reference to it in results section.

2. Molecular weight marker is missing in Suppl. Fig. 2B.

Answer

Has been added.

3. Figure 4 and 5 are not correctly referenced in the Results section.

Answer

Corrected.

Referee #2 (Comments on Novelty/Model System for Author):

In this manuscript, the authors describe experiments demonstrating that the interaction between TREM2 and A β oligomers can block interaction with another ligand and that AD-associated TREM2 variants bind A β with equivalent affinity but show loss of function in terms of signaling and A β internalization.

...

While the paper is timely and important this reviewer has several reservations regarding the data and

their presentation.

1. It is difficult to evaluate the impact of the findings presented. While the binding experiments are convincing, they do not reveal a groundbreaking idea, as Abeta binding by TREM2 has been proposed by others as well (most recently, by Zhao et al., *Neuron* 2018; Zhong et al., *Mol Neurodegen* 2018). The concept that TREM2 variants affect the activation of downstream cascades, is interesting, is based on in vitro experiments alone, and is apparently inconsistent with the study of Kober et al. *J Mol Biol* 2016, showing that in R47H knock-in mice, TREM2 binding to Abeta oligomers is reduced. Similarly, the authors cite their own study (Song et al. *J Exp Med* 2018) in the Discussion as contradicting their present data, defining their own data as "a conundrum", but without attempting to provide clarity by explaining these contradictory results.

Answer

We agree with the reviewer about the novelty. However, our manuscript was submitted to EMBO Mol Med before the publication of these papers. EMBO Molecular Medicine has a "scooping protection" policy, whereby similar findings that are published by others during review or revision are not a criterion for rejection.

We modified our discussion based on these two recent papers.

Kober et al. and Song et al. did not show a direct interaction between TREM2 and amyloid- β . Song et al. showed that TREM2 interacts with neurons and plaques during amyloid- β accumulation and R47H impairs this interaction.

We show no difference in the binding of A β with TREM2 or the AD mutants in vitro. The in vitro experiment has its own limitation. Our result shows that TREM2 AD mutants reduced the signaling activity of NFAT and the amyloid uptake, which tends to result a loss of function. The conclusion of our study is not in contradiction with the report from Song et al.

2. It is unclear how to interpret the NFAT assay. The authors should discuss this issue. Perhaps the author can address the effects of the purportedly protective T69K variant, as comparative NFAT assays, which may lend a stronger support to the partial loss-of-function hypothesis.

Answer

We added a short explanation in the results section to clarify the assay principle, which was previously published in *Cell*. We are not sure regarding the "variant" T69K. This variant T69K cannot be found on pubmed or in the TREM2 data base of mutations on Alzforum. However, we used the putatively associated FTD mutation T96K instead.

3. In general, the paper is not carefully written. Besides recurrent typos (and occasional poorly-constructed sentences), there are obvious portions of text in the Results section that would fit better in either the Methods (as in the first paragraph) or Discussion (as at the end of the final paragraph) sections.

Answer

We apologize if the manuscript was not as polished as it could have been. We corrected the manuscript for language typos.

2nd Editorial Decision

04 September 2018

Thank you for the submission of your revised manuscript to EMBO Molecular Medicine. We have now received the enclosed report from the referee who was asked to re-assess it. As you will see the reviewer is now supportive and I am pleased to inform you that we will be able to accept your manuscript pending minor editorial amendments.

Please submit your revised manuscript within two weeks. I look forward to seeing a revised form of your manuscript as soon as possible.

***** Reviewer's comments *****

Referee #2 (Remarks for Author):

The authors adequately addressed this referees comments.

Corresponding Author Name: Todd E Golde

Journal Submitted to: EMBO MOL MED

Manuscript Number: EMM-2018-09027-V2